# Modeling Competition between Countries in the Development of Arctic Resources

**Pavel Konyukhovskiy** [1] , **Victoria Holodkova** [1] **and Aleksander Titov** [2,*]

[1] Department of Economic Cybernetics, Saint Petersburg State University, St. Petersburg 199034, Russia; aura4004@gmail.com (P.K.); holodkova_v@mail.ru (V.H.)
[2] Information Center of SPbPU in Madrid, 28001 Madrid, Spain
\* Correspondence: aleksandertitov085@gmail.com; Tel.: +7-921-757-51-96

**Abstract:** The article is devoted to the approaches that can be applied in the distribution of Arctic resources between the main reference countries of this region. The objective economic nature of the problems that arise in this region makes it possible to characterize them as a competition of claims for a limited and potentially dynamically changing resource. At a formal level, this problem has a general nature and it is typical for many areas of modern economy. At the same time, it is impossible to deny its specifics, which imposes significant restrictions on possible methods of solution. In recent years, problems in the sphere of interstate cooperation under conditions of limited resources have significantly increased. In such a situation, scientific and practical research in the field of mechanisms for regulating the relations between the parties (economic entities) becomes interesting. In analyzing the mechanisms of distribution of limited resources, one can use the theory of cooperative games, mathematical models of resource rationing, as well as works on the study of problems of equitable distribution (s.c. Fair Divisions). In the framework of such tasks, the range of applicants for limited resources can be limited to countries or regions directly adjacent. The process can be include of "external players" who have sufficient investment potential. The subsequent development and analysis of the problems of regulating intercountry interaction are associated with mathematical formalization. Such formalization presupposes a description of the situation of competitive interaction between countries in the form of a stochastic cooperative game. An analysis of possible concepts for the solution of this game will lead to meaningful conclusions about specific schemes (mechanisms) of rationing.

**Keywords:** Fair Divisions; cooperative games; resource allocation; cooperative stochastic games

## 1. Introduction

In modern times of active development of the world economic system and the exhaustion of natural resources, great importance is given to the regulation of resource competition in the market. These resources are strategically important for each country, and every country is interested in forming a large stock of such resources [1–5].

An important factor in the emergence of resource competition is the limitations and exhaustiveness of the various natural resources that humanity uses to meet its growing needs. The active development of various areas of the economy, including the development of technology and technology makes mankind's use of increasingly difficult resources extracted, take care of the future development and the growing consumption of resources of the country [6].

Today's economic situation is characterized by an active growth of consumption in all sectors. However, it is impossible to ignore the complex problems caused by the objective limitation of global

resources, as well as the fact that the growth of consumption inevitably leads to a rise in the cost of resources in the long term [7–13].

Many natural resources can be concentrated in a large natural area. Such territories may belong to two or more countries. In addition, there are a large number of fossil resources whose localization is difficult to determine. Such resources may be located deep underground. There are a large number of methods and approaches to both detecting such resources and predicting the amount of resources discovered [14–18]. However, with the discovery of natural resources, which are indivisible and located in several countries, the challenge is the identification of such areas. With the development of methods for the study of natural resources, the task becomes more critical, because the discovery of resources often occurs in areas difficult to access.

In recent years, the problems in the sphere of interstate cooperation have intensified. The main factor is the difficulty of determining the boundaries. In this regard, the issues of solving the problems of competitive interaction between countries in the distribution of influence on the use of limited resources are of great importance.

Worldwide, there are a large number of different types of zones in which there is a large number of inter-country conflicts related to the definition of ownership of a natural resource.

One of the main areas of global cooperation of countries where there is resource competition is the Arctic zone. The emergence of competitive interests in this zone is associated with the need to develop new gas and oil fields, the wealth of this zone with natural resources, and the lack of clearly defined boundaries of resource wealth between the countries. There are also technological and climatic reasons. Scientific and technological progress provides new opportunities for work in the Arctic. Climate change increases the attractiveness of Arctic projects [19–21].

Most often, researchers understand the Arctic zone as the northern polar region of our planet, which includes also the extreme coastlines of the continents Eurasia and North America. In addition, the Arctic zone falls in the Arctic Ocean with most of the islands, and some adjacent parts of the Atlantic and Pacific Ocean [21–27].

The Arctic has territories and zones of eight countries, including Russia, Canada, USA, Norway, Denmark, Finland, Sweden and Iceland. The total area of the Arctic zone is estimated at 22–27 million square kilometers (according to various estimates).

Currently, assessments of hidden natural resources that may be located in this natural area are being actively carried out. According to various data, the Arctic zone may contain up to 12 billion tons of oil reserves, as well as natural gas up to 1500 trillions of tons. However, these estimates are preliminary, obtained by expert means and can be seriously corrected in the future [28–34].

In connection with the development of technologies, there are many new opportunities for the extraction of minerals, the extraction of which was previously too costly. A large number of natural resources are concentrated in the Barents, Pechora and Kara seas [35–41].

The Arctic zone is also rich in reserves of copper-nickel ores, tin, gold and other metals.

Therefore, many countries would like to have an impact on such strategic mineral resources. For Russia, this interest is special, due to the fact that the Russian Federation has the largest length of borders with the Arctic [42–47].

There are many situations in which it is possible to use the algorithms of rational resource allocation.

As specific examples of complex ("acute") situations of rationing, we can cite the negotiation process between oil exporting countries, the intensification of competitive confrontation in the gas market, and interaction and competition on a range of problems related to access to resources of the Arctic and Antarctic [47].

The prospects for economic growth in the near future are obvious. At the same time, attention can be drawn to the signs of exhaustion of economic development strategies. It should be noted that the solution of such problems is possible for a wide variety of resources: energy, mineral, environmental quotas, labor, financial and other resources.

One of the important areas of research in this area is the theory of optimal resource allocation.

By the mid-20th century, theoretical economists had largely failed to use mathematical approaches and justification. But mathematical research actively continued, and brilliant results were achieved. In 1975, L. Kantorovich and T.C. Koopmans [48] were awarded the Nobel prize in Economics "for contributions to the theory of optimum allocation of resources".

Currently, the basis of the theory of optimal resource allocation is the method of linear programming, which was first justified by L. Kantorovich. The main method is to maximize limited resources.

For the first time works on the basics of the theory of optimal resource allocation were published in 1939, "Mathematical methods of organization and production planning". In it, Kantorovich proposed a fundamentally new class of extreme problems with constraints, suggesting an effective method for solving them. This theory has a number of disadvantages, namely, the need for a clear quantitative definition of all parameters. However, at present, the understanding of resource allocation issues has gone forward.

The specificity of the problems of rationalization and the so-called fair distribution in modern conditions is manifested in the universalization of their nature. At the moment, the world economic system is facing the need to develop some common, mutually acceptable, agreed and understandable rules and principles for solving the problems of distribution, taking into account the effects of reducing the overall resource base over the long term.

In such a situation, scientific and practical research in the field of mechanisms of regulation of relations between the parties (economic entities) that claim to participate in projects for the development of the Arctic region becomes interesting. In the analysis of the mechanisms of distribution of Arctic resources, one can use the theory of cooperative games, mathematical models of resource rationing (E. Moulin [2]), as well as works on the study of problems of equitable distribution (fair divisions).

One of the important directions of research in this area is the theory of optimal allocation of resources.

Within the framework of such tasks, the range of applicants for Arctic resources can be limited to countries directly adjacent to the Arctic (USA, Canada, Russia, Norway, Denmark-Greenland). It is likely that "external players" with sufficient investment potential (China, Japan, Korea) will be included in the process.

The subsequent development and analysis of the problems of regulation of inter-country cooperation in the Arctic region are associated with mathematical formalization. This formalization involves the description of the situation of competitive interaction of countries in the form of a stochastic cooperative game. The analysis of possible concepts of the solution of this game will allow to come to meaningful conclusions about specific schemes (mechanisms) of rationing [49,50].

The specificity of the problems of rationing [6,7] and equitable distribution in modern conditions is manifested in the universalization of their nature. The world economic system is now faced with the need to develop some common, mutually acceptable, coherent and understandable rules and principles for dealing with distribution tasks that take into account the effects of a reduction in the overall resource base over the long term.

Research in the field of Fair Divisions, the starting point for which largely became the work of E. Moline (E. Mullen), has become one of the most actively developing areas of mathematical Economics in recent years.

The classical objects of study in the works on Fair Divisions were schemes (models) of rationing resources between a finite set of agents (participants of the distribution procedure, players). The results of distribution in the simplest situation are determined by the structure of requirements (applications, claims) of agents and the volume of the distributed resource. To date, the axiomatic properties of a wide range of rationing schemes [3] have been studied in depth: proportional, uniform wins, uniform losses, "random priority", "Talmud".

The fundamental result of modern rationing theory is proof of the relationship between the rules of distribution (according to a particular scheme) and the solution of some "derivative" cooperative

game. Despite the obvious importance and content of these results, their application in practice faces a number of serious difficulties [51–53]. One of the most significant is the ignoring of uncertainty in such models. Indeed, in many cases it is extremely difficult to describe the requirements of the players (applicants for the resource) in the form of uniquely determined values. One of the ways to overcome this difficulty is connected to the transition from deterministic cooperative games to the stochastic cooperative games.

In this context, the stochastic cooperative games are games in which the values of the characteristic function are assumed to be random variables with known distribution functions. At the initial stages of qualitative research, hypotheses about the normal distribution are quite acceptable. The arguments in favor of this hypothesis are based on traditional probability-theoretic approaches (Limit Theorems of Probability Theory).

In this study, it is proposed to modify the traditional situations of rationing resources, introducing a hypothesis that the requirements of the participants are described by random variables. Connected with the corresponding "derivative" of cooperative games will allow us to obtain substantial new results and their interpretations [54–57].

Another promising area of research in this area is related to the identification of stable links between the specifics of the resource and the specific scheme of rationing.

The tenth years of the XXI century, or, to be more precise, the period of development of the world economic system that came after the crisis of 2008–2009 marked the beginning of a new stage of resource competition.

As before, the tasks of economic growth are declared as some unconditional and indisputable priority. At the same time, signs of exhaustion of growth-based economic development strategies are becoming more and more visible.

Consumption growth has become an indisputable and fundamental driver of economic growth. The principle "to consume more next year than this year" has taken the place of the leading paradigm of modern society [58–61]. However, it is impossible to ignore the complex problems caused by the objectively limited global resources, as well as the fact that the growth of consumption inevitably leads to a rise in labor costs in the long term and negatively affects the labor market.

These circumstances can be seen as an argument in favour of the notion that, in the foreseeable future, we will witness a dramatic increase in the role of resource rationalization.

In this case, the authors quite consciously use such a general concept as "resources" since the relevance and urgency of the problems of the fair distribution will increase for a wide variety of types of resources: energy, environmental quotas, labor, in particular, the last "reserves of cheap labor" [62,63].

The negotiation process between the oil exporting countries, aggravation of competitive confrontation in the gas market, interaction and rivalry over a complex of problems of access to the resources of the Arctic and Antarctica can be cited as concrete examples of complex situations of rationalization.

## 2. Materials and Methods

Problems of rationing become relevant in situations where the volume of claims by interested parties relative to a certain resource exceeds its volume.

The focus of this study is on the practical aspects of applying various methods (schemes) for solving problems of distribution-limited resources. In the context of Arctic region problems, it will primarily be mineral minerals, oil and gas.

*Models (Schemes) of Rationing*

Recall also the main provisions of the theory of rationing models (fair distribution), which are supposed to be used in the subsequent presentation. The simplest problem of rationing is defined by a set (set of three factors)

$$(I, t, y) \tag{1}$$

where

$I = \{1, \dots, n\}$—a set of agents (players) ($i \in I$);
$t$—the actual amount of the resource that agents demand;
$y = (y_1, y_2, \dots, y_i, \dots, y_n)$—vector of the space $R^n$, defining the individual requirements of the agents (for the distributed resource).

It is assumed that $t \geq 0$ and $y_i \geq 0$ ($\forall\, i \in I$). The solution of the problem of rationing (method of rationing) is called a vector $x \in R^n$, such that:

$$x = r(I, t, y) \tag{2}$$

$$0 \leq x_i \leq y_i, \ (\forall\, i \in I), \ \sum_{i \in I} x_i = t \tag{3}$$

From a substantive point of view, research on the analysis and comparison of the properties possessed by certain possible methods of rationing is of natural interest. In terms of economic and socio-economic applications, this makes it possible to formalize the notion of requirements for "good", "fair", "mutually acceptable", "objectively indisputable" distribution (satisfaction of competitive claims).

Formally, the requirements for good distribution are postulated in the form of axioms or rules. These rules determine the principles by which the distribution of a resource is formed. Obviously, for different types of resources, the rules can be different. The most famous of the axioms in which such requirements are formulated are presented below.

From a substantive point of view, studies on the analysis and comparison of the properties possessed by certain possible methods of rationing are of natural interest. Within the framework of the formulated tasks (1)–(3) it is necessary to find a total distribution of claims, which in economic and socio-economic terms would allow us to justify the best solution to satisfy competitive claims.

The basic requirements for such distribution can be formulated in the form of basic axioms [8].

*Equal Treatment of Equals* (ETE)—agents who present equal demands obtain an equal share.

*Symmetry* (SYM). This axiom (requirement) assumes that the rationing method $x = pr(I, t, y)$ is a symmetric function. Recall that a function $f(x_1, x_2, \dots, x_j, \dots, x_n)$ is called symmetric if, for any permutation of its variables, the value of the function does not change:

$$f(x_1, x_2, \dots, x_j, \dots, x_n) = f\left(x_{i_1}, x_{i_2}, \dots, x_{i_j}, \dots, x_{i_n}\right).$$

*No Advantageous Reallocation* (NAR)—the change in demand between agents within a group (coalition) $S$ does not change the total share received by the coalition.

*Irrelevance of Reallocations* (IR)—shifts (changes) of claims within a certain group (coalition) $S$ should not affect the shares received by other agents not affected by the redistribution.

*Independence of Merging and Splitting* (IMS)—independence from mergers and divisions of agents, i.e., as a result of the formation of a new agent when merging, its share should be equal to the sum of the shares of merged agents. When dividing one agent into several, its share without balance should be distributed among them.

*Decomposition* (DEC). The performance of this axiom implies the identity of the results of the method of rationing in the case of its application to all possible partitions of the set $I$.

*Zero Consistency* (ZC)—deleting a member with zero demand should not affect the shares received by other members.

As a rule, it is not possible to offer a distribution scheme in which all of the listed requirements would be fulfilled. However, by selecting a set of criteria that applies to a particular type of resource, you can create a model.

Next, we will consider several methods for solving the problem of allocation of limited resources or the problem of equitable distribution. Each of the methods satisfies a number of the proposed rules or axioms.

The simplest method for solving the problem (1) is a *proportional method* that distributes the resource between the agents according to the demand:

$$x = pr(I, t, y) = \frac{t}{y_I} \cdot y \tag{4}$$

or

$$x_i = pr(I, t, y)_i = \frac{t}{y_I} \cdot y_i \tag{5}$$

where $y_I = \sum_{i \in I} y_i$. It is assumed that if is $y_I = 0$ then $x = 0$.

It is easy to prove that the proportional method satisfies the properties NAR, IR, IMS, DEC [31].

Pay attention to the following meaningful moment. NAR or IR axioms are quite reasonable and natural for many of the rationing problems. However, in the case of competition problems for Arctic resources, their feasibility raises serious doubts. This is primarily due to the need to take into account the complex of historical, political and geopolitical factors that are inevitably present in inter-country interaction.

Next in popularity after the proportional method are methods that focus on leveling winnings and losses of agents. There are methods of uniform gains and uniform losses. The *method of uniform gains* (UG) specifies the shares of agents as:

$$x_i = ug_i(I, t, y) = \min\{\lambda, y_i\} \tag{6}$$

where $\lambda$ is found from the equation

$$\sum_{i \in I} \min\{\lambda, y_i\} = t \tag{7}$$

In essence, the UG method implements the principle of leveling gains in relation to the initial requirements, which are obtained by the results of the distribution of the participants of the division.

Another method opposite to the method of uniform gains is *the uniform loss method* (UL). The method (UL) defines the shares received by agents as:

$$x_i = ul_i(I, t, y) = \max\{0, y_i - \mu\} \tag{8}$$

where $\mu$ is found from the equation

$$\sum_{i \in I} \max\{0, y_i - \mu\} = t \tag{9}$$

The UG method implements the principle of leveling "net losses" of players ($y_i - x_i$).

Like the proportional method, the UG and UL methods satisfy axioms of relative ranking:

$$y_i \leq y_j \Rightarrow x_i \leq x_j \tag{10}$$

$$y_i \leq y_j \Rightarrow (y_i - x_i) \leq (y_i - x_j) \tag{11}$$

These methods differ significantly in terms of ranking relative wins. It is easy to show that the UG method satisfies the axiom of progressivity, according to which, if $0 < y_i \leq y_j$, then

$$\frac{x_j}{y_j} \leq \frac{x_i}{y_i} \tag{12}$$

and the UL method satisfies the regressivity axiom:

$$0 < y_i \leq y_j \Rightarrow \frac{x_i}{y_i} \leq \frac{x_j}{y_j} \tag{13}$$

At the same time, it should be noted that the UG does not satisfy the regressivity axiom and UG does not satisfy to progressivity axiom.

Important properties of the ration methods are postulated by axioms of the composition.

In accordance with the axiom top songs UC (*UC—upper composition*)

$$0 \leq t \leq t' \leq y_I \Rightarrow r(I, t, y) = r(I, t, r(I, t', y)) \tag{14}$$

In terms of content, the property of the upper composition is relevant in the case when the actual amount of allocated resources ($t$) was less than initially expected $t'$. UC means the opportunity to review the originally planned optimistic proportion ($r(I, t', y)$) as the requirements of agents in the distribution of actually realized resources $t$.

According to the axiom of the *lower composition* LC:

$$0 \leq t' \leq t \leq y_I \Rightarrow r(I, t, y) = r(I, t', y) + r(I, t - t', y - r(I, t', y)) \tag{15}$$

LC is related to situations when the initial distribution of pessimistic volume ($t'$) occurs, and in the case of additive ($t - t'$), it is distributed according to the same method, taking into account previously satisfied requirements ($y - r(I, t', y)$).

E. Moulin [31], in particular, proved that the method of uniform wins (UG) is characterized by axioms LC and ZC, and the method of uniform losses (UL)—axioms UC and ZC.

Let us introduce the notation. $\pi = (\pi_1, \pi_2, \ldots, \pi_i, \ldots, \pi_n)$—is a permutation of the set of numbers of agents $I$.

The random priority method is defined as:

$$x = \frac{1}{n!} \sum_{\pi \in \Pi} prio(\pi)(I, t, y) \tag{16}$$

where $\Pi$—is the set of all possible permutations of the set $I$.

The distribution rule $\sum_{\pi \in \Pi} prio(\pi)(I, t, y)$ implies that agent $\pi_1$ has the highest priority, agent $\pi_1$—next, etc. Resources are allocated in the descending order of priority (in accordance with the requirements of $y_i$). Thus, in the general case, the requirement of some $\pi_k$-th agent is satisfied in part, the requirements of the agents with the numbers $\pi_{k+1}$, $\pi_{k+2}$, etc. remain unmet.

*The Talmud Method*

$$x = tal(I, t, y) = ug\left(I, \min\left\{t, \frac{y_I}{2}\right\}, \frac{y}{2}\right) + ul\left(I, \max\left\{0, t - \frac{y_I}{2}\right\}, \frac{y}{2}\right) \tag{17}$$

In essence, the Talmud method halves the requirements of the agents and distributes the resource in accordance with the UG-method. In the case that after this the undistributed part of the resource remains, then it is would be distributed according to the UL-method on the basis of half requirements also.

Any of these principles can be implemented in the problem of allocation of scarce resources (oil, gas and other mineral resources) in the Arctic region. However, its application requires to take into account the specific conditions necessary to determine both resources in accordance with the requirements of agents [64].

There is a possibility of implementation of such principles in the construction of models of cooperative games. Cooperative games with transferable utility and characteristic functions can be constructed on the basis of the rationing problem (2):

$$v(S) = \min\{y_s, t\} \tag{18}$$

$$w(S) = \max\{0, t - y_s\} \tag{19}$$

where $S \subset I$ is a coalition of players—possible association of participants of the procedures for the allocation of the resource. $y_s$—sum of demands of participants of the coalition $S$.

Game (18) corresponds to optimistic expectations of participants, game (19), on the contrary–pessimistic.

In this game we will focus on inter-country cooperation in the distribution of rights to the Arctic resources. Perhaps such a distribution is associated with the uncertainty of the boundaries of resources located in the Arctic region.

The theorem [2,32] is valid, according to which the method of random priority distributes resources according to the Shapley-value of games (19), (20); the method of Talmud—in accordance with their—nucleolus.

In accordance with the statement of this theorem, we can express the shares obtained by agents using the random priority method as

$$x_i = \sum_{0 \leq s \leq n-1} \left[ \frac{s! \cdot (n-s-1)!}{n!} \cdot \left( \sum_{S \subset \{I \setminus i\}, |S|=s} (\min\{t, y_{S \bigcup i}\} - \min\{t, y_S\}) \right) \right] \tag{20}$$

where $s = |S|$ is the number of members in coalition $S$ (compare with method (16)).

In turn, on the basis of these methods can be developed methods of resource allocation in the framework of stochastic normalization schemes, similar to the methods of the Talmud and random priority. Further development of the theory of resource allocation is associated with the construction of stochastic cooperative game interaction of countries in the allocation of rights to resources, in which the requirements of agents are defined as random variables $\widetilde{y}_i$ ($i \in I$).

The baseline definition and parameters of a stochastic cooperative game (SCG) is a pair of sets $\Gamma = (I, \widetilde{v})$ where $I = \{1 \ldots n\}$—set of participants; $\widetilde{v}(S)$—random variables with known density functions $P_{\widetilde{v}(s)}(y)$, which are interpreted as income (utility, payoffs), and are received by the corresponding coalitions $S \subset I$.

The solution of this game will allow participants to realize the fair division of a limited resource subject based on the requirements of the participants. In this case, the problem of equitable distribution of Arctic resources among the main applicant countries can be solved. As part of solving the problem of distribution coalitions can be formed, which will receive certain advantages in the distribution of limited economic resources.

One of the most important problems in the practical use of models of rationing is identification of deterministic point values of the agent's requirements $y_i$. Overcoming this problem is possible due to the complexity of the models and the introduction of assumptions about which demands of the agents are random variables $(\widetilde{y}_i)$ with known distribution functions. In fact, the introduction of this premise means the transition to stochastic schemes (models) of rationing [65].

In this case, the characteristic functions of cooperative games, corresponding to stochastic normalization schemes (by analogy with the characteristic functions of games (18) and (19)), will have the form [5]:

$$\widetilde{v}(S) = \min\{\widetilde{y}_s, t\} \tag{21}$$

$$\widetilde{w}(S) = \max\left\{0, t - \widetilde{y}_{I \setminus S}\right\} \tag{22}$$

In turn, on the basis of these methods can be developed methods of resource allocation in the framework of stochastic normalization schemes, similar to the Talmud and random priority methods. This approach, involving a combination of stochastic cooperative games and rationing schemes, is relatively innovative and, from the point of view of the authors, can be quite fruitful.

The application of these approaches provides some insight into the possible distribution of resources among countries. You need to pay attention to the fact that resources of various kinds can be distributed based on different rules. For example, minerals are tied to the territory of occurrence and production of resources; fish stocks migrate, and to determine the volume of their catch in each country, it is advisable to apply quotas for fishing.

## 3. Results and Discussion

In our example, we will consider the analysis of the most important resources for the Arctic zone—the distribution of oil and gas resources. Most countries in the Arctic region have high hopes for the discovery and production of new energy resources on the Arctic shelf.

As already mentioned, the main eight countries claiming resources located in the Arctic are Russia, USA, Canada, Norway, Denmark, Finland, Sweden and Iceland.

Of course, there are countries that can claim such resources do not have direct access to them, such as China.

However, within the framework of this study, we will not affect this area of competition.

The Table 1 below shows the forecast volume of the countries' demand for undeveloped gas reserves in the Arctic region in million cubic meters per year (open source data).

**Table 1.** Volume requirement of the main countries (agents) of the Arctic region.

| Countries | Requirements | Million Cubic Meters per Year |
|---|---|---|
| Russia | $y_1$ | 44.80 |
| USA | $y_2$ | 7.70 |
| Canada | $y_3$ | 1.70 |
| Norway | $y_4$ | 2.00 |
| Denmark | $y_5$ | 0.80 |
| Finland | $y_6$ | 0.05 |
| Sweden | $y_7$ | 1.50 |
| Iceland | $y_8$ | 0.40 |

Based on the formulas proposed above, we consider options for applying various rules of equitable distribution, including the proportional method (see Formulas (4) and (5)), the UG-(6) and (7) and UL-methods (8) and (9) and the Talmud method (17).

The simplest and most convenient approach of this division method is proportional to the requirements. Of course, each particular participant is likely to not get the right amount and the one who has more requirements will be in the worst situation, because its loss in absolute terms can be very significant. This is the main drawback of the method. With all the simplicity and ease of use, this is an essential advantage of this method [66,67].

The calculations set a limit on the expected amount of resources in the required period; this amount (*t*) is equal to 40 million cubic meters per year.

Let us present the results obtained by the PR-method (Formulas (4) and (5)) in Table 2.

**Table 2.** Agent shares for the PR-method.

| Countries | Variable | Result |
|---|---|---|
| Russia | $x_1$ | 30.40 |
| USA | $x_2$ | 5.22 |
| Canada | $x_3$ | 1.15 |
| Norway | $x_4$ | 1.36 |
| Denmark | $x_5$ | 0.54 |
| Finland | $x_6$ | 0.03 |
| Sweden | $x_7$ | 1.02 |
| Iceland | $x_8$ | 0.27 |

According to the results of the calculation, we see that due to the lack of resources, each of the participants receives a proportionally smaller amount of resources than the volume for which he originally claimed.

Table 3 shows the result of the distribution in accordance with the UG-methods (6) and (7). We obtained $\lambda = 25.85$ for these data.

**Table 3.** Agent shares for the UG-method.

| Countries | Variable | Formula | Result |
|-----------|----------|---------|--------|
| Russia | $x_1$ | $\min\{\lambda, y_1\}$ | 25.85 |
| USA | $x_2$ | $\min\{\lambda, y_2\}$ | 7.70 |
| Canada | $x_3$ | $\min\{\lambda, y_3\}$ | 1.70 |
| Norway | $x_4$ | $\min\{\lambda, y_4\}$ | 2.00 |
| Denmark | $x_5$ | $\min\{\lambda, y_5\}$ | 0.80 |
| Finland | $x_6$ | $\min\{\lambda, y_6\}$ | 0.05 |
| Sweden | $x_7$ | $\min\{\lambda, y_7\}$ | 1.50 |
| Iceland | $x_8$ | $\min\{\lambda, y_8\}$ | 0.40 |

It is seen (Table 3) that the result of the distribution by the method of the uniform loss does not coincide with the result of the distribution by the proportional method. In particular, the results are satisfied with more players claiming a smaller volume. Large losses in the implementation of claims for resources will be incurred by those players (countries) that claim a large amount of resources.

Table 4 contains the distribution results in accordance with the UL-method, Formulas (8) and (9). We obtained $\mu = 6.25$ for these data. The result is obtained in which only the first two participants applying for a limited amount of resource will receive a resource volume close to the required volume.

**Table 4.** Agent shares for the UL-method.

| Countries | Variable | Formula | Result |
|-----------|----------|---------|--------|
| Russia | $x_1$ | $\max\{0, y_1 - \mu\}$ | 38.55 |
| USA | $x_2$ | $\max\{0, y_2 - \mu\}$ | 1.45 |
| Canada | $x_3$ | $\max\{0, y_3 - \mu\}$ | 0.00 |
| Norway | $x_4$ | $\max\{0, y_4 - \mu\}$ | 0.00 |
| Denmark | $x_5$ | $\max\{0, y_5 - \mu\}$ | 0.00 |
| Finland | $x_6$ | $\max\{0, y_6 - \mu\}$ | 0.00 |
| Sweden | $x_7$ | $\max\{0, y_7 - \mu\}$ | 0.00 |
| Iceland | $x_8$ | $\max\{0, y_8 - \mu\}$ | 0.00 |

Table 5 contains information on interim calculations for the TAL-method (distribution in accordance with the halved requirements of agents in accordance UG and UL) and the final results for the TAL-method distribution (see method (17)).

A summary of all indicators is presented in Table 6. One of the main conclusions that can be drawn from this table is that the trends of all distributions in some way correspond to the trend of distribution of requirements for limited types of resources, in this case, oil and gas resources of the Arctic region.

**Table 5.** Agent shares for the TAL-method.

| Countries | Formula | Result | Formula | Result | Sum Result |
|---|---|---|---|---|---|
| Russia | $\min\{\lambda, y_1/2\}$ | 22.40 | $\max\{0, y_1/2 - \mu\}$ | 10.52 | 32.92 |
| USA | $\min\{\lambda, y_2/2\}$ | 3.85 | $\max\{0, y_2/2 - \mu\}$ | 0.00 | 3.85 |
| Canada | $\min\{\lambda, y_3/2\}$ | 0.85 | $\max\{0, y_3/2 - \mu\}$ | 0.00 | 0.85 |
| Norway | $\min\{\lambda, y_4/2\}$ | 1.00 | $\max\{0, y_4/2 - \mu\}$ | 0.00 | 1.00 |
| Denmark | $\min\{\lambda, y_5/2\}$ | 0.40 | $\max\{0, y_5/2 - \mu\}$ | 0.00 | 0.40 |
| Finland | $\min\{\lambda, y_6/2\}$ | 0.03 | $\max\{0, y_6/2 - \mu\}$ | 0.00 | 0.03 |
| Sweden | $\min\{\lambda, y_7/2\}$ | 0.75 | $\max\{0, y_7/2 - \mu\}$ | 0.00 | 0.75 |
| Iceland | $\min\{\lambda, y_8/2\}$ | 0.20 | $\max\{0, y_8/2 - \mu\}$ | 0.00 | 0.20 |

**Table 6.** Distribution of requirements according to different approaches (rationing methods).

| Countries | Requirements Millions of Cubic Meters (per year) | | PR | UG | UL | TAL |
|---|---|---|---|---|---|---|
| Russia | $y_1$ | 44.8 | 30.40 | 25.85 | 38.55 | 32.92 |
| USA | $y_2$ | 7.7 | 5.22 | 7.70 | 1.45 | 3.85 |
| Canada | $y_3$ | 1.7 | 1.15 | 1.70 | 0.00 | 0.85 |
| Norway | $y_4$ | 2 | 1.36 | 2.00 | 0.00 | 1.00 |
| Denmark | $y_5$ | 0.8 | 0.54 | 0.80 | 0.00 | 0.40 |
| Finland | $y_6$ | 0.05 | 0.03 | 0.05 | 0.00 | 0.03 |
| Sweden | $y_7$ | 1.5 | 1.02 | 1.50 | 0.00 | 0.75 |
| Iceland | $y_8$ | 0.4 | 0.27 | 0.40 | 0.00 | 0.20 |
| | | 58.95 | 40.00 | 40.00 | 40.00 | 40.00 |
| | $t$ | 40 | | | | |
| | $\mu$ | 25.85 | | | | |
| | $\lambda$ | 6.25 | | | | |

The total volume of claims of the countries reaches 58.95 million cubic meters and exceeds the projected volume of extracted resources by 18.95 million cubic meters, and this is due to the need to establish an equitable distribution of claims for resources from different countries applying for them.

Table 6 presents the results of the calculations. Two countries (with the largest needs), in any form of distribution, receive some resources (more than half of the needs)—Table 6. At the same time, countries applying for a much smaller amount of resources (in proportional terms) do not receive anything using the method of uniform losses [59,66–68]. In other cases, the level of needs for these countries will be small.

This difference in allocation depends on the method you choose when you allocate the resource.

As a result of the analysis, we can identify three main groups of country-applicants for Arctic resources: first, Russia and the United States; the third group includes all the other 6 countries, the total requirements of which do not exceed the requirements of the second largest participant in the division (USA).

The next stage of analysis, which can be carried out based on the results, is the formation of a graphical representation of the distribution of limited Arctic resources. Graphical representation (Figure 1) shows the distribution of resources by country of applicants presented in Tables 6 and 7.

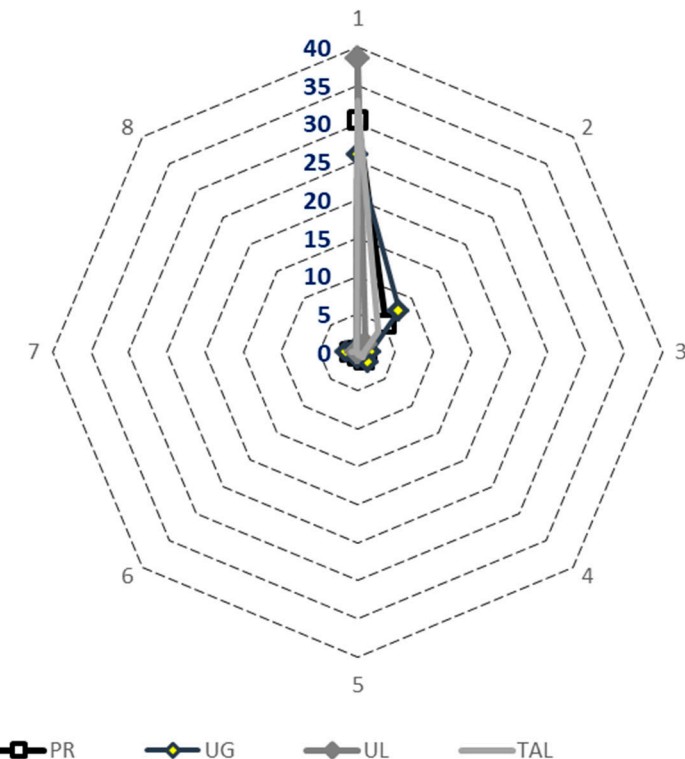

**Figure 1.** Graphic visualization of solutions to the problem of the Arctic resource distribution, obtained using different rationing methods.

**Table 7.** The distribution of participants according to groups of applicant countries for resources.

| Countries | Distribution, Millions of Cubic Meters | PR | UG | UL | TAL |
|---|---|---|---|---|---|
| $y_1$ | 44.80 | 30.40 | 25.85 | 38.55 | 32.92 |
| $y_2$ | 7.70 | 5.22 | 7.70 | 1.45 | 3.85 |
| $y_3 + \cdots + y_8$ | 6.45 | 4.38 | 6.45 | 0 | 3.23 |

The chart shows eight axes according to the number of candidate countries based on the allocation of resources [69]. For each axis of the graph, the volumes allocated as a result of the division are postponed for each country of the resource.

The analysis of these rationing methods and their applicability to specific situations of competition for resources is an extremely interesting and informative direction for future research.

Formation of the task distribution of Arctic resources will be reduced to the following task:

Many agents (players) are the main contenders for resources in the Arctic, as we have already mentioned, Russia, Canada, USA, Norway, Denmark, Finland, Sweden and Iceland. In addition, some countries may have joint investors in the region; consequently, the number of participants can be adjusted to reduce and increase. Note that to solve the game with fewer participants will require fewer resources.

The estimated resource volume is indicated by -t. By resource is understood-oil, gas, mineral raw materials. For each type of resource, the participants of the game have their own requirements for distribution.

Thus, it is necessary to determine the vector $y = (y_1, y_2, \ldots, y_i, \ldots, y_n)$—the volume of requirements imposed by agents on the distributed resource [68–72]. The evaluation of the vector $x$ is one of the most difficult and contested quantities in the problem. One of the options assessment may be the assessment on the basis of expert estimations of the world's leading experts, as well as on the

basis of existing trends in the development of economic relations. Other options for evaluation may be based on the existing needs of a given resource in a country. Other options suitable for a particular resource may also be considered.

To solve the problem—create a schema for the equitable distribution of the resource, you must define a vector $x_i$—volume resource that will be able to claim one or the other country (or that player).

The use of the above algorithms implies the need for a detailed analysis of the balance of forces of the game participants (distribution of Arctic resources). Formation of the task for each type of resource requires a detailed analysis of the participants and characteristics of the resource.

## 4. Conclusions

The results obtained by each method can demonstrate the best distribution of requirements across the selected resource types. The participants of the game (countries or their associations) will receive the volume of resource (oil, gas and other resources) that meets the specified requirements (on the basis of the chosen rule). Thus, the solution of the problem will be an equitable distribution of resources under the given economic conditions.

Solving the problem of distribution of rights to the Arctic resources under the current economic conditions is an urgent task. None of the researchers can say exactly what rights a particular country or region has to a particular type of resource. Nevertheless, the proposed game models allow us to illustrate the diverse situations that may arise under the conditions of inter-country competition. These models will help to examine in detail the impact of countries' positions on the results of the actual distribution of rights to limited resources in the future.

Note that the need for such a situation may arise in various sectors of the economy; however, the situation of resource allocation in the Arctic region is relevant. Currently, the solution of such problems for resources in the disputed territories, which became available in connection with the development of new technologies and improvement of technology, is becoming increasingly important.

As a conclusion, it should be noted that the purpose of interaction between the state today is to create conditions for sustainable operation, the preservation of solvency and the formation of comfortable conditions for the existence of citizens. For this domestic purpose, most states are doing well, but their position in the external environment has a serious impact on the internal policy of the state.

The concepts of a monopolar world today do not lose their relevance. The immediate centre of this monopoly remains a big question. If a few years ago the absolute leader was recognized by the United States, today their positions have significantly deteriorated.

The question of maintaining a monopolar world remains a big question, the concept of a multi-polar world seems more obvious, where the most likely contenders for the role of the leading powers are several—the US and China, Russia and Europe, which can seriously resist them.

What coalitions can be formed? At present, it is obvious that America is striving to create a multipolar world, but having lost its position, gives way to other participants, such as China, which has long maintained neutrality, but with a serious strengthening of the negative influence of the United States and attempts to influence its policy, decided to apply the policy of creating a coalition with other powers, in particular with Russia. Such a game (when it comes to the theory of games) makes it clear that the center, though retained by the United States, already significantly begins to outweigh the united coalition (Russia and China).

Today, Russia, even under the most optimal circumstances, will not be able to become the center of the monopolar world due to the weakness of its influence on all emerging coalitions.

At the same time, in any situation, each of the participants (USA, Europe, Russia, China) will seek to create a coalition with other participants to strengthen their positions to create a stable situation for themselves.

These trends also have an impact on the achievement of specific objectives in the area of equitable distribution of resources, including in the area of claims by states that do not have territorial points of contact with resource sources.

**Author Contributions:** The authors contributed equally to this work. P.K. carried out the formation of the model, as well as the construction and selection of rules for determining the distribution. In the implementation of practical calculations, they obtained results by the method of the Talmud. V.H. carried out the selection of data and practical calculations in the implementation of the assessment of the distribution of countries in the Arctic region using the proportional method, the method of uniform income and a single method of loss. A.T. carried out the systematization of the material and the construction of graphical visualization of solutions to the problem of distribution of resources of the Arctic, obtained by various methods of rationing.

**Funding:** This research was funded by the Russian Science Foundation (Project No. 14-38-00009, the program-targeted management of the Russian Arctic zone development, Peter the Great St. Petersburg Polytechnic University) and the Russian Science Foundation, 109992, Russian Federation, Moscow, http://rscf.ru/.

**Acknowledgments:** The paper is based on research carried out with the financial support of a grant from the Russian Science Foundation (Project No. 14-38-00009, the program-targeted management of the Russian Arctic zone development) and Peter the Great St. Petersburg Polytechnic University.

**Conflicts of Interest:** The authors declare no conflict of interest.

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
