# Peer review of "Modeling Competition between Countries in the Development of Arctic Resources"

_resources, doi:10.3390/resources8010049_

Round 1
Reviewer 1 Report
In "Modeling of competition between countries in the development of Arctic resources" authors study an interesting and vibrant problem, and they put forward innovative and highly useful results, which I am sure might inspire future research along similar lines.
The following comments should nevertheless be taken into account if a revision will be granted.
1) In the introduction, when referring to research related to this subject in general, and in particular to sustainability and the betterment of our lives through innovative research, the review Saving human lives: What complexity science and information systems can contribute, J. Stat. Phys. 158, 735-781 (2015) is closely related and should be useful to the readers for a more informative introduction.
2) It would also improve the paper if the figure captions would be made more self-contained. In addition to briefly stating what is shown, one could also consider a sentence or two saying what is the main message of each figure.
3) In general, figures from the data would be useful too. The tables are quite extensive, with lots of numbers, and it is somewhat difficult for the reader to make sense of it all. The authors should reconsider the presentation of the results.
4) In terms of recent application of stochastic games and game theory in general to social and ecological problems, research Excessive abundance of common resources deters social responsibility, Sci. Rep. 4, 4161 (2014), and the review Phase transitions in models of human cooperation, Phys. Lett. A 380, 2803-2808 (2016) are a recent and a more modern approach to the well-mixed formulism applied in this paper. In general, I am not fully convinced we are not able to do better in terms of the application of the theory the specific problem at hand. It is fine the authors go from some very early works, but the applications of game theory and stochastic games in the past decade have been so many, with so many methodological advances, including taking into account the network of contacts and many other ecological constraints, that it is difficult to fully appreciate what the authors have done because the approach inevitably requires simplificating assumptions on so many levels of the studied problem.
Author Response
Thanks for the interesting and thoughtful comments on the proposed theme. We will certainly take them into account in our future work.
Most of the comments were taken into account. We have tried to take the comments in the English part.
As for the comment on l. 200: the function $x$ is denoted by $r$, and the same function on l. 219 by $pr$. Function r() - general function. Pr() - function for the proportional method.
Once again, we еxpress our gratitude for the careful study of our article and making significant proposals.

Reviewer 2 Report
The paper is devoted to fair division problem in the concrete context of Arctic resources. Though the theoretical results of the paper are almost negligible (standard formulas are taken and numbers substituted), I feel in this case that the publication would be really very useful in principle, because of the obvious practical importance of the question. The author made a very good job by reviewing the literature, collecting the data and making impressive amount of calculations that may serve as a starting point for more advanced theoretical analysis and modeling. Unfortunately I have to add that the quality of exposition is far from being satisfactory at present, especially bad is the language, but some notations and formulas are of concern. Therefore the serious revision is necessary. The quality clearly deteriorates, as the exposition progresses. First two pages are mostly OK.
Below are some concrete points (not exhaustive).
l. 83: Language "in this situation there are many situations"
l. 91: Language: senseless sentence
l. 97: "Kantorovich and the American Professor Koopman". Was Kantorovich not a professor? Or is it stressed that he was stateless as compared with American professor?
l. 100 "by" is missing.
One can definitely shorten the whole presentations as it has lots of repetition, some explicit and some hidden so-to-say. For instance, the explicit repetition are lines 109-111 and 131-134. Also, say, when giving two different tables obtained by different methods, what is the point to stress specifically that they are different (l. 394). Of course they are.
l. 151: "deterministic cooperative games in stochastic" This is a very strange term.
l. 155: Why Gaussian distributions are " quite acceptable" in this context?
l. 186: Language "we will call the rule the distribution"
why on l. 200 the function $x$ is denoted by $r$, and the same function on l. 219 by $pr$?
Why the numbers of formulas randomly oscillate along the lines, while their standard place should be
at the fixed right end position?
l. 221: mess with the notation for permutation
l. 235: Why is it obvious?
l. 238: "we propose". Really? It looks like you are describing and using standard methods, and not propose something new.
Why there are 2 formulas with the number (2)?
l. 246: something wrong with English
l. 251: bad English
l. 254: "method ... is methods" terrible English
l. 258 index is missing in the formula
l. 278 senseless sentence
l. 291 " denote by ... is a permutation" terrible English
l. 302 Language, possibly should be "uses half of", but surely not "makes half each".
l. 315: Again wrong numbering of formulas.
l. 326, 344 Again bad English
l. 390: "method-the method"
And so on an so forth.
Author Response

(The authors gave the same response as above.)

Round 2
Reviewer 1 Report
The authors have revised their manuscript comprehensively and with love to detail. I warmly recommend publication in present form.
Reviewer 2 Report
The authors made a serious revision. The paper is much better now. I can recommend publication in the present form.